# A rigorous calculation of the Feynman and Wheeler Scalar Fields propagators in the ADS / CFT correspondence using distribution theory

A. Plastino[1,3,4], M.C.Rocca[1,2,3]

[1] Departamento de Física, Universidad Nacional de La Plata,

[2] Departamento de Matemática, Universidad Nacional de La Plata,

[3] Consejo Nacional de Investigaciones Científicas y Tecnológicas

(IFLP-CCT-CONICET)-C. C. 727, 1900 La Plata - Argentina

[4] SThAR - EPFL, Lausanne, Switzerland

February 8, 2019

## Abstract

By appeal to Distribution Theory we discuss in rigorous fashion, without appealing to **any conjecture** (as usually done by other authors), the boundary-bulk propagators for the scalar field, both in the non-massive and massive cases. These calculations, new in the literature as far as we know, are carried out in two instances: (i) when the boundary is a Euclidean space and (ii) when it is of Minkowskian nature. In this last case we compute also three propagators: Feynman's, Anti-Feynman's, and Wheeler's (half advanced plus half retarded). For an operator corresponding to scalar field we obtain the two points correlations functions in the three instances above mentioned
**PACS**11.25.-w, 04.60.Cf, 02.30.Sa, 03.65.Db.
**KEYWORDS** Distribution theory, ADS/CFT correspondence; Boundary-bulk propagators; Feynman's propagators, Wheeler's propagators.

# 1 Introduction

Propagators and correlators are one of the essential tools to work, for example, in Quantum Field Theory (QFT) and String Theory (ST). In particular, in formulating the correspondence ADS/CFT (Anti-de Sitter/ Conformal Field Theory)/ This correspondence was established by Maldacena [1] in 1998 and is universally regarded as a very useful model for many purposes. The bibliography on this subject, for scalar fields, is quite extensive. We give here just a small part of it in [2, 3, 4, 5, 6, 7, 8, 9]. For a more complete bibliography the reader is directed to the report [10] One of the ADS/CFT correspondence's prescriptions (see [2]) will allow us to evaluate the correlators on the boundary of ADS space The first boundary-bulk propagator was calculated by Witten a few months after the appearance of [1], entitled *Anti de Sitter space and holography*. In this case the boundary is an Euclidean space [2, 3].

By appeal to distribution theory, in this work we evaluate instead the boundary-bulk propagators for the case in which the boundary is a Minkowskian space. Distribution theory is not a tool employed by AdS/CFT practitioners, as far as we know.

In such regard, remark that some attempts have been made before in [11, 12, 13]. The only previous (and almost ) attempt to try to calculate boundary-bulk propagators in the Minkowskian boundary for the Anti-de Sitter space [in the ADS/CFT correspondence] was made by Son and Starinets (SS) in 2002 [14]. However, SS needed to formulate a conjecture that we show here to become unnecessary if one uses the full distributions-theory of type $S'$ (of Schwartz). This important work was entitled "Minkowski-space correlators in AdS/CFT correspondence: recipe and applications". We must also mention the work of Freedman et al. [21], in which the authors deal with the case of a Euclidean boundary. Freedman, however, did not treat the case of a Minkowskian boundary, at least in the way that Son and Starinets did. Note that we make full use of distribution theory. This does not entail, of course, a simple $i\epsilon$ prescription, but a much more elaborate treatment, that has not been performed before in this field. Let us remark, as this is an important point for us, that in this paper we do not evaluate renormalized correlation functions. We will do that in a forthcoming paper using the method given in [21].

Thus, in the present effort we evaluate, without any conjecture, the boundary-bulk propagators corresponding to the following three cases i) Feynman, ii)

Anti-Feynman, and iii) Wheeler. We do this both, for a massless scenario and for the massive ones, (a scalar field involved). Later we calculate the two points correlators (TPC) for operators corresponding to this scalar field in the three instances previously mentioned. We clarify that in this paper we do not evaluate the renormalized TPC. We will do that in a next paper using the method given in [21].

In the present effort we demonstrate that the Feynman propagator must be a function of $\rho + i0$ (see below for the notation) in momentum space, and therefore a function of $x^2 - i0$ in configuration space. We show that something similar happens with the Anti-Feynman propagator. *For the first time ever*, we calculate the Wheeler's propagator as well. *Note that, until the 90's, the only field propagators that had been calculated were Anti-de Sitter (spatial) ones.*

The paper is organized as follows: Section 2 deals with with the Euclidean case. In it, the three different propagators referred to above can not be distinguished (neither in the massive nor in the massless instances).

In section 3 we tackle similar scenarios as those of section 2, but now in Minkowski's space, where the three propagators can be distinguished.

In section 4 we compute in Euclidean space the TPC for a scalar operator corresponding to a scalar field via Witten's prescription

In section 5 we generalize the calculations of section 4 to Minkowski's space. We obtain in this fashion the two-points correlations functions corresponding to the three different propagators of our list above.

Finally, some conclusions are drawn in section 6.

# 2   Euclidean Case

## 2.1   Massless Scalar Field Propagator

The Klein-Gordon equation in $ADS_{\nu+1}$ for the scalar field $\phi(z, \vec{x})$ reads, in Poincare coordinates,

$$z^2 \partial_z^2 \phi(z, \vec{x}) + (1 - \nu) z \partial_z \phi(z, \vec{x}) + z^2 \nabla^2 \phi(z, \vec{x}) - \Delta(\Delta - \nu) \phi(z, \vec{x}) = 0, \ (2.1)$$

where $\Delta(\Delta - \nu) \geq 0$ plays the role of $m^2$. We exclude tachyons from of this treatment. Here $\Delta$ is the conformal dimension, $\nu$ the boundary's dimension, and $\vec{x}$ their coordinates. The Fourier transform in the variables $\vec{x}$ of the field

$\phi(z, \vec{x})$ is

$$\hat{\phi}(z, \vec{k}) = \int \phi(z, \vec{x}) e^{i\vec{k}\cdot\vec{x}} d^\nu x. \tag{2.2}$$

Using (2.2), (2.1) takes the form

$$z^2 \partial_z^2 \hat{\phi}(z, \vec{k}) + (1 - \nu) z \partial_z \hat{\phi}(z, \vec{k}) - [z^2 k^2 + \Delta(\Delta - \nu)] \hat{\phi}(z, \vec{k}) = 0. \tag{2.3}$$

We analyze now the massless case given by $\Delta = 0, \nu$. For it we have the motion equation

$$z^2 \partial_z^2 \hat{\phi}(z, \vec{k}) + (1 - \nu) z \partial_z \hat{\phi}(z, \vec{k}) - z^2 k^2 \hat{\phi}(z, \vec{k}) = 0, \tag{2.4}$$

or equivalently (for $z \neq 0$)

$$\partial_z^2 \hat{\phi}(z, \vec{k}) + \frac{(1 - \nu)}{z} \partial_z \hat{\phi}(z, \vec{k}) - k^2 \hat{\phi}(z, \vec{k}) = 0. \tag{2.5}$$

In the variable $z$, this equation is of the Bessel type (see [18])

$$F''(z) + \frac{(1 - 2\alpha)}{z} F'(z) - \left[ k^2 + \frac{\mu^2 - \alpha^2}{z^2} \right] F(z) = 0 \tag{2.6}$$

The pertinent solution (that does not diverge when the argument tends to infinity) is

$$F(z) = z^\alpha \mathcal{K}_\mu(kz). \tag{2.7}$$

Thus, the solution of (2.5) becomes

$$\hat{\phi}(z, k) = z^{\frac{\nu}{2}} \mathcal{K}_{\frac{\nu}{2}}(kz). \tag{2.8}$$

One easily verifies that, for infinitesimal $z$ [18],

$$\mathcal{K}_{\frac{\nu}{2}}(kz) = \frac{2^{\frac{\nu}{2}-1} \Gamma\left(\frac{\nu}{2}\right)}{(kz)^{\frac{\nu}{2}}} + O\left((kz)^{-\frac{\nu}{2}+2}\right), \tag{2.9}$$

and therefore

$$\lim_{z \to 0} z^{\frac{\nu}{2}} \mathcal{K}_{\frac{\nu}{2}}(kz) = \frac{2^{\frac{\nu}{2}-1} \Gamma\left(\frac{\nu}{2}\right)}{k^{\frac{\nu}{2}}}. \tag{2.10}$$

In other words, the solution is regular at the origin and vanishes at infinity (in the variable $z$). Accordingly, we have, for the field in the bulk, the solution

$$\phi(z, \vec{x}) = \frac{z^{\frac{\nu}{2}}}{(2\pi)^\nu} \int a(\vec{k}) \mathcal{K}_{\frac{\nu}{2}}(kz) e^{-i\vec{k}\cdot\vec{x}} d^\nu k. \tag{2.11}$$

This solution must reduce itself to the field $\phi_0(x)$ on the boundary, so that

$$\phi(0, \vec{x}) = \phi_0(\vec{x}) = \frac{2^{\frac{\nu}{2}-1}\Gamma\left(\frac{\nu}{2}\right)}{(2\pi)^{\nu}} \int a(\vec{k}) k^{-\frac{\nu}{2}} e^{-i\vec{k}\cdot\vec{x}} d^{\nu}k =$$

$$\frac{1}{(2\pi)^{\nu}} \int \hat{\phi}_0(\vec{k}) e^{-i\vec{k}\cdot\vec{x}} d^{\nu}k. \tag{2.12}$$

From this last equation we can obtain $a(k)$ as a function of $\hat{\phi}_0$ and then write

$$\phi(z, \vec{x}) = \frac{z^{\frac{\nu}{2}} 2^{1-\frac{\nu}{2}}}{(2\pi)^{\nu}\Gamma\left(\frac{\nu}{2}\right)} \int k^{\frac{\nu}{2}} \mathcal{K}_{\frac{\nu}{2}}(kz) \hat{\phi}_0(\vec{k}) e^{-i\vec{k}\cdot\vec{x}} d^{\nu}k, \tag{2.13}$$

or, equivalently,

$$\phi(z, \vec{x}) = \frac{z^{\frac{\nu}{2}} 2^{1-\frac{\nu}{2}}}{(2\pi)^{\nu}\Gamma\left(\frac{\nu}{2}\right)} \iint k^{\frac{\nu}{2}} \mathcal{K}_{\frac{\nu}{2}}(kz) \phi_0(\vec{x'}) e^{-i\vec{k}\cdot(\vec{x}-\vec{x'})} d^{\nu}k d^{\nu}x'. \tag{2.14}$$

From (2.14) we then obtain an expression of the boundary-bulk propagator

$$K(z, \vec{x} - \vec{x'}) = \frac{z^{\frac{\nu}{2}} 2^{1-\frac{\nu}{2}}}{(2\pi)^{\nu}\Gamma\left(\frac{\nu}{2}\right)} \int k^{\frac{\nu}{2}} \mathcal{K}_{\frac{\nu}{2}}(kz) e^{-i\vec{k}\cdot(\vec{x}-\vec{x'})} d^{\nu}k. \tag{2.15}$$

To carry out the integration in the variable $k$ we appeal to the expressions for the Fourier transform and its inverse obtained by Bochner [17]. For the Fourier transform we have

$$\hat{f}(k) = \int f(\vec{x}) e^{i\vec{k}\cdot\vec{x}} d^{\nu}x = \frac{(2\pi)^{\frac{\nu}{2}}}{k^{\frac{\nu}{2}-1}} \int_0^{\infty} r^{\frac{\nu}{2}} \mathcal{J}_{\frac{\nu}{2}-1}(kr) f(r) dr, \tag{2.16}$$

and for its inverse

$$f(r) = \frac{1}{(2\pi)^{\nu}} \int \hat{f}(\vec{k}) e^{-i\vec{k}\cdot\vec{x}} d^{\nu}k = \frac{1}{(2\pi)^{\frac{\nu}{2}} r^{\frac{\nu}{2}-1}} \int_0^{\infty} k^{\frac{\nu}{2}} \mathcal{J}_{\frac{\nu}{2}-1}(kr) \hat{f}(k) dk. \tag{2.17}$$

Using these relations we have now

$$K(z, \vec{x} - \vec{x'}) = \frac{z^{\frac{\nu}{2}} 2^{1-\frac{\nu}{2}}}{(2\pi)^{\nu}\Gamma\left(\frac{\nu}{2}\right)} \frac{(2\pi)^{\frac{\nu}{2}}}{|\vec{x} - \vec{x'}|^{\frac{\nu}{2}-1}} \int_0^{\infty} k^{\nu} \mathcal{K}_{\frac{\nu}{2}}(kz) \mathcal{J}_{\frac{\nu}{2}-1}(k|\vec{x} - \vec{x'}|) dk. \tag{2.18}$$

So as to evaluate the last integral we appeal to a result from [18]

$$\int_0^\infty x^{\mu+\nu+1} \mathcal{K}_\nu(bx) \mathcal{J}_\mu(ax) dx = 2^{\mu+\nu} a^\mu b^\nu \frac{\Gamma(\mu+\nu+1)}{(a^2+b^2)^{\mu+\nu+1}}, \qquad (2.19)$$

Our deduction follows a different, simpler and complete path than that of [2]. Our approach also has a didactic utility.

$$K(z, \vec{x} - \vec{x'}) = \frac{\Gamma(\nu)}{\pi^{\frac{\nu}{2}} \Gamma\left(\frac{\nu}{2}\right)} \left[ \frac{z}{z^2 + (\vec{x} - \vec{x'})^2} \right]^\nu, \qquad (2.20)$$

which leads to

$$\phi(z, \vec{x}) = \int K(z, \vec{x} - \vec{x'}) \phi_0(\vec{x'}) d^\nu x', \qquad (2.21)$$

an expression that, in turn, leads to

$$\lim_{z \to 0} K(z, \vec{x} - \vec{x'}) = \delta(\vec{x} - \vec{x'}). \qquad (2.22)$$

## 2.2 Massive Field Propagator

We now consider the massive case $\Delta \neq 0, \nu$. The equation of motion for this case reads

$$z^2 \partial_z^2 \hat{\phi}(z, \vec{k}) + (1-\nu) z \partial_z \hat{\phi}(z, \vec{k}) - [z^2 k^2 + \Delta(\Delta - \nu)] \hat{\phi}(z, \vec{k}) = 0, \qquad (2.23)$$

or equivalently,

$$\partial_z^2 \hat{\phi}(z, \vec{k}) + \frac{(1-\nu)}{z} \partial_z \hat{\phi}(z, \vec{k}) - \left[ k^2 + \frac{\Delta(\Delta - \nu)}{z^2} \right] \hat{\phi}(z, \vec{k}) = 0. \qquad (2.24)$$

The solution for this last equation is

$$\hat{\phi}(z, k) = z^{\frac{\nu}{2}} \mathcal{K}_\mu(kz), \qquad (2.25)$$

with

$$\mu = \pm \sqrt{\frac{\nu^2}{4} + \Delta(\Delta - \nu)}. \qquad (2.26)$$

Since $\mathcal{K}_\mu(z) = \mathcal{K}_{-\mu}(z)$, we select for $\mu$ in (2.26) the plus sign. We have then

$$\phi(z, \vec{x}) = \frac{z^{\frac{\nu}{2}}}{(2\pi)^\nu} \int a(\vec{k})\mathcal{K}_\mu(kz)e^{-i\vec{k}\cdot\vec{x}}d^\nu k. \tag{2.27}$$

For $\Delta \neq 0$, this solution is not regular at the origin. To overcome this problem we select

$$\phi(\epsilon, \vec{x}) = \phi_\epsilon(\vec{x}) = \frac{\epsilon^{\frac{\nu}{2}}}{(2\pi)^\nu} \int a(\vec{k})\mathcal{K}_\mu(k\epsilon)e^{-i\vec{k}\cdot\vec{x}}d^\nu k =$$

$$\frac{1}{(2\pi)^\nu} \int \hat{\phi}_\epsilon(\vec{k})e^{-i\vec{k}\cdot\vec{x}}d^\nu k, \tag{2.28}$$

where $\epsilon$ is infinitesimal. From (2.28) we have then

$$a(\vec{k}) = \frac{\hat{\phi}_\epsilon(\vec{k})}{\epsilon^{\frac{\nu}{2}}\mathcal{K}_\mu(k\epsilon)}. \tag{2.29}$$

Replacing the result of (2.29) into (2.27) we obtain

$$\phi(z, \vec{x}) = \frac{1}{(2\pi)^\nu} \left(\frac{z}{\epsilon}\right)^{\frac{\nu}{2}} \int \frac{\mathcal{K}_\mu(kz)}{\mathcal{K}_\mu(k\epsilon)}\hat{\phi}_\epsilon(\vec{k})e^{-i\vec{k}\cdot\vec{x}}d^\nu k, \tag{2.30}$$

or similarly,

$$\phi(z, \vec{x}) = \frac{1}{(2\pi)^\nu} \left(\frac{z}{\epsilon}\right)^{\frac{\nu}{2}} \int \int \frac{\mathcal{K}_\mu(kz)}{\mathcal{K}_\mu(k\epsilon)}\phi_\epsilon(\vec{x'})e^{-i\vec{k}\cdot(\vec{x}-\vec{x'})}d^\nu k d^\nu x'. \tag{2.31}$$

From this last equation we see that the propagator is

$$K_m(z, \vec{x} - \vec{x'}) = \frac{1}{(2\pi)^\nu} \left(\frac{z}{\epsilon}\right)^{\frac{\nu}{2}} \int \frac{\mathcal{K}_\mu(kz)}{\mathcal{K}_\mu(k\epsilon)}e^{-i\vec{k}\cdot(\vec{x}-\vec{x'})}d^\nu k. \tag{2.32}$$

As a consequence we can write

$$\phi(z, \vec{x}) = \int K_m(z, \vec{x} - \vec{x'})\phi_\epsilon(\vec{x'})d^\nu x'. \tag{2.33}$$

From (2.33) we immediately gather that

$$K_m(\epsilon, \vec{x} - \vec{x'}) = \delta(\vec{x} - \vec{x'}). \tag{2.34}$$

## 2.3 Approximate Massive Field Propagator

We are now going to discuss a non-valid approach for the function $\mathcal{K}(k\epsilon)$. The issue here is that, although $\epsilon$ is infinitesimal, it can not adopt the $0-$value. As $k$ is an unbounded variable, when $k \to \infty$, we have $k\epsilon \to \infty$. Notice first that

$$\mathcal{K}_\mu(k\epsilon) = \frac{2^{\mu-1}\Gamma(\mu)}{(k\epsilon)^\mu} + O((k\epsilon)^{2-\mu}). \tag{2.35}$$

We now make the approximation

$$\mathcal{K}_\mu(k\epsilon) = \frac{2^{\mu-1}\Gamma(\mu)}{(k\epsilon)^\mu}. \tag{2.36}$$

From (2.32) we obtain an approximation for the propagator $K$ that we shall call $M$. We have then

$$M_m(z, \vec{x} - \vec{x}') = \frac{1}{(2\pi)^\nu} \left(\frac{z}{\epsilon}\right)^{\frac{\nu}{2}} \frac{\epsilon^\mu}{2^{\mu-1}\Gamma(\mu)} \int k^\mu \mathcal{K}_\mu(kz) e^{-i\vec{k}\cdot(\vec{x}-\vec{x}')} d^\nu k. \tag{2.37}$$

Using again the Bochner formula we arrive at

$$\int k^\mu \mathcal{K}_\mu(kz) e^{-i\vec{k}\cdot(\vec{x}-\vec{x}')} d^\nu k = \frac{(2\pi)^{\frac{\nu}{2}}}{|\vec{x}-\vec{x}'|^{\frac{\nu}{2}-1}} \int_0^\infty k^{\mu+\frac{\nu}{2}} \mathcal{K}_\mu(kz) \mathcal{J}_{\frac{\nu}{2}-1}(k|\vec{x}-\vec{x}'|) dk. \tag{2.38}$$

By recourse to (2.19) we then have

$$M_m(z, \vec{x} - \vec{x}') = \frac{\epsilon^{\mu-\frac{\nu}{2}}}{\pi^{\frac{\nu}{2}}} \frac{\Gamma\left(\mu + \frac{\nu}{2}\right)}{\Gamma(\mu)} \left[\frac{z}{z^2 + (\vec{x}-\vec{x}')^2}\right]^{\mu+\frac{\nu}{2}}. \tag{2.39}$$

We now define

$$\gamma = \frac{\nu}{2} + \mu = \frac{\nu}{2} + \sqrt{\frac{\nu^2}{4} + \Delta(\Delta - \nu)}, \tag{2.40}$$

so that we can write

$$M_m(z, \vec{x} - \vec{x}') = \frac{\epsilon^{\gamma-\nu}}{\pi^{\frac{\nu}{2}}} \frac{\Gamma(\gamma)}{\Gamma\left(\gamma - \frac{\nu}{2}\right)} \left[\frac{z}{z^2 + (\vec{x}-\vec{x}')^2}\right]^\gamma. \tag{2.41}$$

We now realize that, by construction,

$$M_m(\epsilon, \vec{x} - \vec{x}') \neq \delta(\vec{x} - \vec{x}'), \tag{2.42}$$

and define
$$N_m(z, \vec{x} - \vec{x}') = M_m(z, \vec{x} - \vec{x}')\epsilon^{\nu - \gamma}, \tag{2.43}$$

which allows us to write for $N_m$ the expression

$$N_m(z, \vec{x} - \vec{x}') = \frac{1}{\pi^{\frac{\nu}{2}}} \frac{\Gamma(\gamma)}{\Gamma\left(\gamma - \frac{\nu}{2}\right)} \left[\frac{z}{z^2 + (\vec{x} - \vec{x}')^2}\right]^{\gamma}. \tag{2.44}$$

Therefore, we have constructively proved that

$$\lim_{z \to 0} N_m(z, \vec{x} - \vec{x}') \neq \delta(\vec{x} - \vec{x}'). \tag{2.45}$$

Note that (2.44) is indeed the well known expression for the boundary-bulk propagator for a scalar field in configuration space. However, this expression can only be used as an approximation to the propagator $K$ when $\mu \cong \frac{\nu}{2}$.

# 3  Minkowskian Case

## 3.1  Massless Field Propagator

Let is now deal with the case in which the boundary of the $ADS_{\nu+1}$ is the $\nu$-dimensional Minkowskian space. In the massless case the field-equation is

$$z^2 \partial_z^2 \hat{\phi}(z, k) + (1 - \nu)z\partial_z \hat{\phi}(z, k) + z^2 k^2 \hat{\phi}(z, k) = 0, \tag{3.1}$$

where $k^2 = k_0^2 - \vec{k}^2 = \rho$. Thus, we can write

$$z^2 \partial_z^2 \hat{\phi}(z, \rho) + (1 - \nu)z\partial_z \hat{\phi}(z, \rho) + z^2 \rho \hat{\phi}(z, \rho) = 0, \tag{3.2}$$

or, rewriting this last equation,

$$z^2 \partial_z^2 \hat{\phi}(z, \rho) + (1 - \nu)z\partial_z \hat{\phi}(z, \rho) - z^2 \left[\mp i(\rho \pm i0)^{\frac{1}{2}}\right]^2 \hat{\phi}(z, \rho) = 0. \tag{3.3}$$

The distribution $(\rho \pm i0)^{\lambda}$ is defined as (see reference [16])

$$(\rho \pm i0)^{\lambda} = \rho_+^{\lambda} + e^{\pm i\pi\lambda}\rho_-^{\lambda}, \tag{3.4}$$

and can be cast in terms of $H(x)$, the Heaviside step function [16]. We recast now (3.3) in the form of a Bessel equation

$$\partial_z^2 \hat{\phi}(z, \rho) + \frac{(1 - \nu)}{z}\partial_z \hat{\phi}(z, \rho) - \left[\mp i(\rho \pm i0)^{\frac{1}{2}}\right]^2 \hat{\phi}(z, \rho) = 0. \tag{3.5}$$

The solution of this equation that is i) regular at the origin and 2) vanishes for $\rho \to \infty$, becomes

$$\hat{\phi}(z,k) = z^{\frac{\nu}{2}} \mathcal{K}_{\frac{\nu}{2}}[\mp i(\rho \pm i0)^{\frac{1}{2}} z]. \tag{3.6}$$

One must take into account that $\lim_{k \to \infty} e^{ikx} = 0$ (see below in this section and [19]).

$$\mathcal{K}_{\frac{\nu}{2}}[\mp i(\rho \pm i0)^{\frac{1}{2}} z] = \frac{2^{\frac{\nu}{2}-1} \Gamma\left(\frac{\nu}{2}\right)}{[\mp i(\rho \pm i0)^{\frac{1}{2}} z]^{\frac{\nu}{2}}} + O\left([\mp i(\rho \pm i0)^{\frac{1}{2}} z]^{-\frac{\nu}{2}+2}\right). \tag{3.7}$$

We have then

$$\phi_{\mp}(z,x) = \frac{z^{\frac{\nu}{2}}}{(2\pi)^{\nu}} \int a(\vec{k}) \mathcal{K}_{\frac{\nu}{2}}[\mp i(\rho \pm i0)^{\frac{1}{2}} z] e^{-ik \cdot x} d^{\nu}k =$$

$$\int \hat{\phi}(z,k) e^{ik \cdot x} d^{\nu}k. \tag{3.8}$$

From this last equation we deduce that

$$\phi_{\mp}(z,x) = \frac{z^{\frac{\nu}{2}} 2^{1-\frac{\nu}{2}}}{(2\pi)^{\nu} \Gamma\left(\frac{\nu}{2}\right)} \int [\mp i(\rho \pm i0)^{\frac{1}{2}}]^{\frac{\nu}{2}} \mathcal{K}_{\frac{\nu}{2}}[\mp i(\rho \pm i0)^{\frac{1}{2}} z] \hat{\phi}_0(k) e^{-ik \cdot x} d^{\nu}k, \tag{3.9}$$

or, equivalently,

$$\phi_{\mp}(z,x) = \frac{z^{\frac{\nu}{2}} 2^{1-\frac{\nu}{2}}}{(2\pi)^{\nu} \Gamma\left(\frac{\nu}{2}\right)} \int \int [\mp i(\rho \pm i0)^{\frac{1}{2}}]^{\frac{\nu}{2}} \mathcal{K}_{\frac{\nu}{2}}[\mp i(\rho \pm i0)^{\frac{1}{2}} z] \times$$

$$\phi_0(x^{'}) e^{-ik \cdot (x-x^{'})} d^{\nu}k d^{\nu}x^{'}. \tag{3.10}$$

The ensuing propagator becomes then

$$K_{\mp}(z, x-x^{'}) = \frac{z^{\frac{\nu}{2}} 2^{1-\frac{\nu}{2}}}{(2\pi)^{\nu} \Gamma\left(\frac{\nu}{2}\right)} \int [\mp i(\rho \pm i0)^{\frac{1}{2}}]^{\frac{\nu}{2}} \mathcal{K}_{\frac{\nu}{2}}[\mp i(\rho \pm i0)^{\frac{1}{2}} z] e^{-i\vec{k} \cdot (x-x^{'})} d^{\nu}k. \tag{3.11}$$

Thus, the corresponding Feynman's propagator is

$$K_F(z, x-x^{'}) = \frac{z^{\frac{\nu}{2}} 2^{1-\frac{\nu}{2}}}{(2\pi)^{\nu} \Gamma\left(\frac{\nu}{2}\right)} \int [-i(\rho + i0)^{\frac{1}{2}}]^{\frac{\nu}{2}} \mathcal{K}_{\frac{\nu}{2}}[-i(\rho + i0)^{\frac{1}{2}} z] e^{-i\vec{k} \cdot (x-x^{'})} d^{\nu}k. \tag{3.12}$$

Note that the Feynman propagator is a function of $\rho + i0$, as it should. For the anti-Feynman propagator we have instead

$$K_{AF}(z, x - x') = \frac{z^{\frac{\nu}{2}} 2^{1-\frac{\nu}{2}}}{(2\pi)^\nu \Gamma\left(\frac{\nu}{2}\right)} \int [i(\rho - i0)^{\frac{1}{2}}]^{\frac{\nu}{2}} \mathcal{K}_{\frac{\nu}{2}}[i(\rho - i0)^{\frac{1}{2}} z] e^{-i\vec{k}\cdot(x-x')} d^\nu k.$$

(3.13)

The expression for the Wheeler's propagator is:

$$W(z, x - x') = \frac{1}{2}[K_F(z, x - x') + K_{AF}(z, x - x')].$$

(3.14)

Using the relations

$$K_F(z, \rho) = \frac{z^{\frac{\nu}{2}} 2^{1-\frac{\nu}{2}}}{\Gamma\left(\frac{\nu}{2}\right)} [-i(\rho + i0)^{\frac{1}{2}}]^{\frac{\nu}{2}} \mathcal{K}_{\frac{\nu}{2}}[-i(\rho + i0)^{\frac{1}{2}} z],$$

(3.15)

and

$$K_{AF}(z, \rho) = \frac{z^{\frac{\nu}{2}} 2^{1-\frac{\nu}{2}}}{\Gamma\left(\frac{\nu}{2}\right)} [i(\rho - i0)^{\frac{1}{2}}]^{\frac{\nu}{2}} \mathcal{K}_{\frac{\nu}{2}}[i(\rho - i0)^{\frac{1}{2}} z],$$

(3.16)

we can define, as usual, the retarded propagator

$$K_R(z, \rho) = H(k^0) K_F(z, \rho) + H(-k^0) K_{AF}(z, \rho),$$

(3.17)

and the advanced propagator

$$K_A(z, \rho) = H(k^0) K_{AF}(z, \rho) + H(-k^0) K_F(z, \rho).$$

(3.18)

We are going to show now that $\lim\limits_{k \to \infty} e^{ikx} = 0$ (see [19]) Let $\hat{\phi}$ be a test function belonging to a sub-space $\mathcal{S}$ of Schwartz's one [15, 16]. Its Fourier transform is

$$\phi(k) = \int\limits_{-\infty}^{\infty} \hat{\phi}(x) e^{ikx} dx,$$

(3.19)

where $\phi$ belongs to $\mathcal{S}$. Then one can verify that

$$0 = \lim_{k \to \infty} \phi(k) = \lim_{k \to \infty} \int\limits_{-\infty}^{\infty} \hat{\phi}(x) e^{ikx} dx = \int\limits_{-\infty}^{\infty} \hat{\phi}(x) \lim_{k \to \infty} e^{ikx} dx.$$

(3.20)

As a consequence, we obtain

$$\lim_{k \to \infty} e^{ikx} = 0$$

(3.21)

The Feynman propagator is, according to (3.12,)

$$K_F(z,x) = \frac{z^{\frac{\nu}{2}} 2^{1-\frac{\nu}{2}}}{(2\pi)^\nu \Gamma\left(\frac{\nu}{2}\right)} \int [-i(\rho+i0)^{\frac{1}{2}}]^{\frac{\nu}{2}} \mathcal{K}_{\frac{\nu}{2}}[-i(\rho+i0)^{\frac{1}{2}}z] e^{-ik\cdot x} d^\nu k. \quad (3.22)$$

Since $\mathcal{K}_{\frac{\nu}{2}}$ is exponentially decreasing or oscillating, we can evaluate the integral that defines $K_F$ by means of a Wick rotation over $k_0$. Therefore we have the change of variables $k_0 = ik_{0E}$, $x_0 = ix_{0E}$, $k_E^2 = k_{0E}^2 + \vec{k}^2$, and $x_E^2 = x_{0E}^2 + \vec{x}^2$. Casting the integral that defines the propagator in terms of these new variables, we obtain

$$K_F(z,\vec{x}_E) = \frac{iz^{\frac{\nu}{2}} 2^{1-\frac{\nu}{2}}}{(2\pi)^\nu \Gamma\left(\frac{\nu}{2}\right)} \int k_E^{\frac{\nu}{2}} \mathcal{K}_{\frac{\nu}{2}}(k_E z) e^{-i\vec{k}_E \cdot \vec{x}_E} d^\nu k_E. \quad (3.23)$$

Using Bochner's formula together with (2.19) we have

$$K_F(z,x_E) = \frac{i\Gamma(\nu)}{\pi^{\frac{\nu}{2}} \Gamma\left(\frac{\nu}{2}\right)} \left[\frac{z}{z^2 + x_E^2}\right]^\nu. \quad (3.24)$$

Now, making the change to Minkowskian variables and taking into account that the Fourier transform of a distribution that depends on $\rho - i0$ is a distribution that depends on $x^2 + i0$, we obtain

$$K_F(z,x) = \frac{i\Gamma(\nu)}{\pi^{\frac{\nu}{2}} \Gamma\left(\frac{\nu}{2}\right)} \left[\frac{z}{z^2 - x^2 - i0}\right]^\nu, \quad (3.25)$$

which is the expression of the Feynman propagator in terms of the variables of the configuration space. For the anti-Feynman propagator we analogously find

$$K_{AF}(z,x) = \frac{i\Gamma(\nu)}{\pi^{\frac{\nu}{2}} \Gamma\left(\frac{\nu}{2}\right)} \left[\frac{z}{z^2 - x^2 + i0}\right]^\nu. \quad (3.26)$$

## 3.2 Massive Field Propagator

For the massive case, the field-motion equation is

$$\partial_z^2 \hat{\phi}(z,\rho) + \frac{(1-\nu)}{z} \partial_z \hat{\phi}(z,\rho) - \left\{ \left[\mp i(\rho \pm i0)^{\frac{1}{2}}\right]^2 + \frac{\Delta(\Delta - \nu)}{z^2} \right\} \hat{\phi}(z,\rho) = 0, \quad (3.27)$$

with, again,

$$\mu = \sqrt{\frac{\nu^2}{4} + \Delta(\Delta - \nu)}. \tag{3.28}$$

The pertinent solution is now

$$\hat{\phi}_{\mp}(z, \rho) = z^{\frac{\nu}{2}} \mathcal{K}_\mu[\mp i(\rho \pm i0)^{\frac{1}{2}} z]. \tag{3.29}$$

The field-expression in configuration space is then

$$\phi_{\mp}(z, x) = \frac{z^{\frac{\nu}{2}}}{(2\pi)^\nu} \int a(\vec{k}) \mathcal{K}_\mu[\mp i(\rho \pm i0)^{\frac{1}{2}} z] e^{-ik \cdot x} d^\nu k. \tag{3.30}$$

Once again we choose

$$\phi(\epsilon, x) = \phi_\epsilon(x) = \frac{\epsilon^{\frac{\nu}{2}}}{(2\pi)^\nu} \int a(\vec{k}) \mathcal{K}_\mu[\mp i(\rho \pm i0)^{\frac{1}{2}} \epsilon] e^{-ik \cdot x} d^\nu k =$$

$$\frac{1}{(2\pi)^\nu} \int \hat{\phi}_\epsilon(k) e^{-ik \cdot x} d^\nu k, \tag{3.31}$$

and from (3.23) we obtain

$$a(k) = \frac{\hat{\phi}_\epsilon(k)}{\epsilon^{\frac{\nu}{2}} \mathcal{K}_\mu[\mp i(\rho \pm i0)^{\frac{1}{2}} \epsilon]}. \tag{3.32}$$

We have then the following relation for the solution

$$\phi_{\mp}(z, x) = \frac{1}{(2\pi)^\nu} \left(\frac{z}{\epsilon}\right)^{\frac{\nu}{2}} \int \int \frac{\mathcal{K}_\mu[\mp i(\rho \pm i0)^{\frac{1}{2}} z]}{\mathcal{K}_\mu[\mp i(\rho \pm i0)^{\frac{1}{2}} \epsilon]} \phi_\epsilon(x') e^{-ik \cdot (x - x')} d^\nu k d^\nu x', \tag{3.33}$$

so that the propagator is now

$$K_{m\mp}(z, x - x') = \frac{1}{(2\pi)^\nu} \left(\frac{z}{\epsilon}\right)^{\frac{\nu}{2}} \int \frac{\mathcal{K}_\mu[\mp i(\rho \pm i0)^{\frac{1}{2}} z]}{\mathcal{K}_\mu[\mp i(\rho \pm i0)^{\frac{1}{2}} \epsilon]} e^{-i\vec{k} \cdot (x - x')} d^\nu k. \tag{3.34}$$

The corresponding Feynman's propagator becomes

$$K_{mF}(z, x - x') = \frac{1}{(2\pi)^\nu} \left(\frac{z}{\epsilon}\right)^{\frac{\nu}{2}} \int \frac{\mathcal{K}_\mu[-i(\rho + i0)^{\frac{1}{2}} z]}{\mathcal{K}_\mu[-i(\rho + i0)^{\frac{1}{2}} \epsilon]} e^{-i\vec{k} \cdot (x - x')} d^\nu k, \tag{3.35}$$

and for the anti-Feynman propagator we obtain the expression

$$K_{mAF}(z, x - x') = \frac{1}{(2\pi)^\nu} \left(\frac{z}{\epsilon}\right)^{\frac{\nu}{2}} \int \frac{\mathcal{K}_\mu[i(\rho - i0)^{\frac{1}{2}} z]}{\mathcal{K}_\mu[i(\rho - i0)^{\frac{1}{2}} \epsilon]} e^{-i\vec{k} \cdot (x - x')} d^\nu k. \tag{3.36}$$

Finally, the definition of Wheeler propagators, retarded and advanced, is similar to that of the preceding subsection.

## 3.3 Approximation

We now evaluate in approximate fashion the propagator $\mathcal{K}_\mu[-i(\rho+i0)^{\frac{1}{2}}\epsilon]$

$$\mathcal{K}_\mu[-i(\rho+i0)^{\frac{1}{2}}\epsilon] = \frac{2^{\mu-1}\Gamma(\mu)}{(-i)^\mu(\rho+i0)^{\frac{\mu}{2}}\epsilon^\mu}, \tag{3.37}$$

entailing

$$M_{mF}(z,x) = \frac{z^{\frac{\nu}{2}}}{(2\pi)^\nu} \frac{\epsilon^{\mu-\frac{\nu}{2}}}{2^{\mu-1}\Gamma(\mu)} \int \mathcal{K}_\mu[-i(\rho+i0)^{\frac{1}{2}}z][-i(\rho+i0)^{\frac{1}{2}}]^\mu e^{-ik\cdot x}d^\nu k. \tag{3.38}$$

Effecting again the above Wick's rotation we obtain

$$M_{mF}(z,\vec{x}_E) = \frac{iz^{\frac{\nu}{2}}}{(2\pi)^\nu} \frac{\epsilon^{\mu-\frac{\nu}{2}}}{2^{\mu-1}\Gamma(\mu)} \int k_E^\mu \mathcal{K}_\mu(k_E z)e^{-i\vec{k}_E\cdot\vec{x}_E}d^\nu k_E. \tag{3.39}$$

This integral is evaluated as in the previous cases. One has

$$M_{mF}(z,\vec{x}_E) = \frac{i\epsilon^{\mu-\frac{\nu}{2}}}{\pi^{\frac{\nu}{2}}} \frac{\Gamma\left(\mu-\frac{\nu}{2}\right)}{\Gamma(\mu)} \left(\frac{z}{z^2+x_E^2}\right)^{\mu+\frac{\nu}{2}}. \tag{3.40}$$

Changing variables as above we arrive at

$$M_{mF}(z,x) = \frac{i\epsilon^{\gamma-\nu}}{\pi^{\frac{\nu}{2}}} \frac{\Gamma(\gamma)}{\Gamma\left(\gamma-\frac{\nu}{2}\right)} \left(\frac{z}{z^2-x^2-i0}\right)^\gamma, \tag{3.41}$$

where

$$\gamma = \frac{\nu}{2} + \sqrt{\frac{\nu^2}{4} + \Delta(\Delta-\nu)}. \tag{3.42}$$

Now we return to the inequality

$$M_{mF}(\epsilon,x) \neq \delta(x) \tag{3.43}$$

The following relation is valid for $N_F$

$$N_{mF}(\epsilon,x) = \epsilon^{\nu-\gamma}M_{mF}(\epsilon,x). \tag{3.44}$$

Proceeding in analogous fashion with the Anti-Feynman propagator we obtain the approximation

$$M_{mAF}(z,x) = \frac{i\epsilon^{\gamma-\nu}}{\pi^{\frac{\nu}{2}}} \frac{\Gamma(\gamma)}{\Gamma\left(\gamma-\frac{\nu}{2}\right)} \left(\frac{z}{z^2-x^2+i0}\right)^\gamma \tag{3.45}$$

# 4  Two points correlation functions in Euclidean space

## 4.1  Massless case

To evaluate the two-points correlation function of a scalar operator we use the result obtained in [20]. This is

$$< \mathcal{O}(\vec{x}_1)\mathcal{O}(\vec{x}_2) >= - \int \sqrt{g}\partial_\mu K(y_0, \vec{y} - \vec{x}_1)\partial^\mu K(y_0, \vec{y} - \vec{x}_2)d^{\nu+1}y, \quad (4.1)$$

where $0 \le y_0 = z < \infty$, $y_\mu = x_\mu$, $\mu \neq 0$, and then

$$< \mathcal{O}(\vec{x}_1)\mathcal{O}(\vec{x}_2) >= - \int_{Boundary} \lim_{z \to 0}[z^{1-\nu}K(z, \vec{x} - \vec{x}_1)\partial_z K(z, \vec{x} - \vec{x}_2)]d^\nu x. \quad (4.2)$$

As $\lim_{z \to 0} K(0, \vec{x}_1 - \vec{x}_2) = \delta(\vec{x}_1 - \vec{x}_2)$, we obtain

$$< \mathcal{O}(\vec{x}_1)\mathcal{O}(\vec{x}_2) >= - \lim_{z \to 0}[z^{1-\nu}\partial_z K(z, \vec{x}_1 - \vec{x}_2)]. \quad (4.3)$$

Using now the expression for $K$ given in Eq.(2.20) we have

$$< \mathcal{O}(\vec{x}_1)\mathcal{O}(\vec{x}_2) >= - \frac{\Gamma(\nu+1)}{\pi^{\frac{\nu}{2}}\Gamma\left(\frac{\nu}{2}\right)}\frac{1}{(\vec{x}_1 - \vec{x}_2)^{2\nu}}. \quad (4.4)$$

Accordingly, we have here arrived to the usual, well-known result.

## 4.2  Massive case

For the massive case we obtain similarly

$$< \mathcal{O}(\vec{x}_1)\mathcal{O}(\vec{x}_2) >_m= - \int_{Boundary} [z^{1-\nu}K_m(z, \vec{x} - \vec{x}_1)\partial_z K(z, \vec{x} - \vec{x}_2)]d^\nu x. \quad (4.5)$$

As $K_m(\epsilon, \vec{x} - \vec{x}_1) = \delta(\vec{x} - \vec{x}_1)$ we can write

$$< \mathcal{O}(\vec{x}_1)\mathcal{O}(\vec{x}_2) >_m= - \int \delta(\vec{x} - \vec{x}_1)[z^{1-\nu}\partial_z K(z, \vec{x} - \vec{x}_2)]_{z=\epsilon}d^\nu x. \quad (4.6)$$

Thus we arrive at

$$< \mathcal{O}(\vec{x}_1)\mathcal{O}(\vec{x}_2) >_m = -[z^{1-\nu}\partial_z K_m(z, \vec{x}_1 - \vec{x}_2)]_{z=\epsilon}. \qquad (4.7)$$

Now, we use the expression for $K_m$ given in (2.32) and write

$$< \mathcal{O}(\vec{x}_1)\mathcal{O}(\vec{x}_2) >_m = -\frac{\epsilon^{1-\nu}}{(2\pi)^\nu}\partial_z\left[\left(\frac{z}{\epsilon}\right)^{\frac{\nu}{2}}\int\frac{\mathcal{K}_\mu(kz)}{\mathcal{K}_\mu(k\epsilon)}e^{-i\vec{k}\cdot(\vec{x}_1-\vec{x}_2)}d^\nu k\right]_{z=\epsilon}, \quad (4.8)$$

or, equivalently,

$$< \mathcal{O}(\vec{x}_1)\mathcal{O}(\vec{x}_2) >_m = -\frac{\epsilon^{1-\frac{3\nu}{2}}}{(2\pi)^\nu}\left[\frac{\nu}{2}z^{\frac{\nu}{2}-1}\int\frac{\mathcal{K}_\mu(kz)}{\mathcal{K}_\mu(k\epsilon)}e^{-i\vec{k}\cdot(\vec{x}_1-\vec{x}_2)}d^\nu k + \right.$$

$$\left. z^{\frac{\nu}{2}}\int k\frac{\mathcal{K}'_\mu(kz)}{\mathcal{K}_\mu(k\epsilon)}e^{-i\vec{k}\cdot(\vec{x}_1-\vec{x}_2)}d^\nu k\right]_{z=\epsilon}. \qquad (4.9)$$

Using now the following result, given in [18],

$$\mathcal{K}'_\mu(z) = -\frac{\mu}{z}\mathcal{K}_\mu + \mathcal{K}_{\mu-1}, \qquad (4.10)$$

we obtain

$$< \mathcal{O}(\vec{x}_1)\mathcal{O}(\vec{x}_2) > = \epsilon^{-\nu}(\mu-\frac{\nu}{2})\delta(\vec{x}_1-\vec{x}_2) - \frac{\epsilon^{1-\nu}}{(2\pi)^\nu}\int k\frac{\mathcal{K}_{\mu-1}(k\epsilon)}{\mathcal{K}_\mu(k\epsilon)}e^{-i\vec{k}\cdot(\vec{x}_1-\vec{x}_2)}d^\nu k.$$
$$(4.11)$$

Note that we have not renormalized the correlation functions. We will do that using the results of [21] in a forthcoming paper.

# 5 Two points correlation functions in Minkowskian space

## 5.1 Massless case

Similarly to the Euclidean case we obtain for the Minkowskian one the result

$$< \mathcal{O}(x_1)\mathcal{O}(x_2) >_F = i\lim_{z\to 0}[z^{1-\nu}\partial_z K_F(z, x_1 - x_2)]. \qquad (5.1)$$

Thus, we obtain for the Feynman's propagator

$$< \mathcal{O}(\vec{x}_1)\mathcal{O}(\vec{x}_2) >_F = -\frac{\Gamma(\nu+1)}{\pi^{\frac{\nu}{2}}\Gamma\left(\frac{\nu}{2}\right)}\frac{1}{[(\vec{x}_1-\vec{x}_2)^2-(x_{10}-x_{20})^2+i0]^{\nu}}, \qquad (5.2)$$

and for Anti-Feynman one

$$< \mathcal{O}(\vec{x}_1)\mathcal{O}(\vec{x}_2) >_{AF} = -\frac{\Gamma(\nu+1)}{\pi^{\frac{\nu}{2}}\Gamma\left(\frac{\nu}{2}\right)}\frac{1}{[(\vec{x}_1-\vec{x}_2)^2-(x_{10}-x_{20})^2-i0]^{\nu}}. \qquad (5.3)$$

## 5.2 Massive case

Again, following the developments of the Euclidean case, we have, for the Minkowskian instance, the two points Feynman's correlator

$$< \mathcal{O}(\vec{x}_1)\mathcal{O}(\vec{x}_2) >_{mF} = i[z^{1-\nu}\partial_z K_{mF}(z, x_1-x_2)]_{z=\epsilon}. \qquad (5.4)$$

Thus, we have

$$< \mathcal{O}(x_1)\mathcal{O}(x_2) >_{mF} = i\frac{\epsilon^{1-\nu}}{(2\pi)^{\nu}}\partial_z\left[\left(\frac{z}{\epsilon}\right)^{\frac{\nu}{2}}\int\frac{\mathcal{K}_{\mu}[-i(\rho+i0)^{\frac{1}{2}}z]}{\mathcal{K}_{\mu}[-i(\rho+i0)^{\frac{1}{2}}\epsilon]}e^{-ik\cdot(x_1-x_2)}d^{\nu}k\right]_{z=\epsilon}, \qquad (5.5)$$

or equivalently,

$$< \mathcal{O}(x_1)\mathcal{O}(x_2) >_{mF} = i\frac{\epsilon^{1-\frac{3\nu}{2}}}{(2\pi)^{\nu}}\left[\frac{\nu}{2}z^{\frac{\nu}{2}-1}\int\frac{\mathcal{K}_{\mu}[-i(\rho+i0)^{\frac{1}{2}}z]}{\mathcal{K}_{\mu}[-i(\rho+i0)^{\frac{1}{2}}\epsilon]}e^{-ik\cdot(x_1-x_2)}d^{\nu}k -\right.$$

$$\left. iz^{\frac{\nu}{2}}\int(\rho+i0)^{\frac{1}{2}}\frac{\mathcal{K}'_{\mu}[-i(\rho+i0)^{\frac{1}{2}}z]}{\mathcal{K}_{\mu}[-i(\rho-i0)^{\frac{1}{2}}\epsilon]}e^{-ik\cdot(x_1-x_2)}d^{\nu}k\right]_{z=\epsilon}. \qquad (5.6)$$

Using again (4.10) we finally obtain

$$< \mathcal{O}(x_1)\mathcal{O}(x_2) >_F = \epsilon^{-\nu}(\mu-\frac{\nu}{2})\delta(x_1-x_2)+$$

$$\frac{\epsilon^{1-\nu}}{(2\pi)^{\nu}}\int(\rho+i0)^{\frac{1}{2}}\frac{\mathcal{K}_{\mu-1}[-i(\rho+i0)^{\frac{1}{2}}\epsilon]}{\mathcal{K}_{\mu}[-i(\rho+i0)^{\frac{1}{2}}\epsilon]}e^{-ik\cdot(x_1-x_2)}d^{\nu}k. \qquad (5.7)$$

For the Anti-Feynman propagator we obtain in analogous fashion

$$< \mathcal{O}(x_1)\mathcal{O}(x_2) >_{mAF} = \epsilon^{-\nu}(\mu-\frac{\nu}{2})\delta(x_1-x_2)-$$

$$\frac{\epsilon^{1-\nu}}{(2\pi)^\nu} \int (\rho - i0)^{\frac{1}{2}} \frac{\mathcal{K}_{\mu-1}[i(\rho - i0)^{\frac{1}{2}}\epsilon]}{\mathcal{K}_\mu[i(\rho - i0)^{\frac{1}{2}}\epsilon]} e^{-ik\cdot(x_1 - x_2)} d^\nu k. \qquad (5.8)$$

Note that we have not renormalized the correlation functions here. We will do so using the results of [21] in a forthcoming paper. We strongly emphasize the fact that holographic renormalization should not be used in this context because it constitutes a non-rigorous mathematical technique.

# 6 Discussion

In this work we have first calculated, without using any conjecture, the boundary-bulk Feynman, Anti-Feynman, and Wheeler propagators for both a massless and a massive scalar field, by recourse to the theory of distributions.
Note that such a conjecture was made in 2002 by Son and Starinets [14], only for the Feynman propagator, and that they did not get a correct result
As further novelties, in the paper we show that, for massive scalar fields, the expression for the boundary-bulk propagator in Euclidean momentum space does not correspond to the expression used in the configuration space, but it is rather a mere approximation.
Subsequently, using the previous results, we have evaluated the correlation functions of scalar operators corresponding to massless and massive scalar fields.
**Unlike the results obtained in [14], with the ones obtained here you can calculate the n-points correlation functions from gravity. This is feasible for a scalar operator when $n$ is an arbitrary natural number.**

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
