# Peer review of "A rigorous calculation of the Feynman and Wheeler Scalar Fields propagators in the ADS / CFT correspondence using distribution theory"

_SciPost Physics_

## Round 2 · Referee Report · Anonymous (Referee 1) · 2019-4-2

Strengths

-

Weaknesses

1- The problem discussed has been already addressed in the literature in more depth than is attempted in the present article.

2- The relevant references are not cited, e.g.:
https://arxiv.org/abs/hep-th/9911182
https://arxiv.org/abs/hep-th/0212072
https://arxiv.org/abs/0805.0150
https://arxiv.org/abs/0812.2909

3- Renormalized 2-point functions are not considered. The unrenormalized 2-point functions are not well defined distributions at coincident points for all scalar masses.

4- Except for the introduction, the article is a compendium of fairly elementary calculations.

Report

Due to the reasons cited above, I deem this article unsuitable for publication.

Requested changes

-

  • validity: low
  • significance: low
  • originality: low
  • clarity: ok
  • formatting: below threshold
  • grammar: reasonable

Author:  Mario Rocca  on 2019-05-10  [id 510]

(in reply to Report 1 on 2019-04-02)

The referee wrote a very poor report that does not address the CONTENTS of our paper.

1- False assertions are made by the referee. For instance, the ADS/CFT Wheeler propagator for a scalar field has been NEVER before discussed in the Literature.

2- The referee gives references that are not related at all to our paper. For instance, the first of them refers to propagators for tensor fields. We deal only with scalar fields.

  • We do cite the main 15 papers on the subject.

  • We also cite Nastase's book.

3-The referee seems to ignore everything about distributions. For instance, he speaks about value of distribution at a point, which is nonsense.. In particular for two point correlation's functions, renormalized or not renormalized.

4- It seems that the referee has not bothered to read our paper in any detail.

With all due respect, we request then a new, more adequate referee.with knowledge on Distribution's Theory.

Mario Rocca

Author:  Mario Rocca  on 2019-04-17  [id 501]

(in reply to Report 1 on 2019-04-02)
Category:
remark

The referee wrote a very poor report that does not address the CONTENTS of our paper.

1- False assertions are made by the referee. For instance, the ADS/CFT Wheeler propagator for a scalar field has been NEVER before discussed in the Literature.

2- The referee gives references that are not related at all to our paper. For instance, the first of them refers to propagators for tensor fields. We deal only with scalar fields.

  • We do cite the main 15 papers on the subject.

  • We also cite Nastase's book.

3-The referee seems to ignore everything about distributions. For instance, he speaks about value of distribution at a point, which is nonsense.. In particular for two point correlation's functions, renormalized or not renormalized.

4- It seems that the referee has not bothered to read our paper in any detail.

---

## Editorial Decision

rejected_or_withdrawn